# The Role of the Interleukin-1 Family in Complications of Prematurity

**DOI:** 10.3390/ijms24032795

**Published:** 2023-02-01

**Authors:** Elys A. Green, Steven P. Garrick, Briana Peterson, Philip J. Berger, Robert Galinsky, Rod W. Hunt, Steven X. Cho, Jane E. Bourke, Marcel F. Nold, Claudia A. Nold-Petry

**Affiliations:** 1Ritchie Centre, Hudson Institute of Medical Research, Melbourne, VIC 3168, Australia; 2Department of Paediatrics, Monash University, Melbourne, VIC 3168, Australia; 3Monash Newborn, Monash Children’s Hospital, Melbourne, VIC 3168, Australia; 4Department of Obstetrics and Gynaecology, Monash University, Melbourne, VIC 3168, Australia; 5Department of Pharmacology, Biomedicine Discovery Institute, Monash University, Melbourne, VIC 3168, Australia

**Keywords:** prematurity, inflammation, bronchopulmonary dysplasia, pulmonary hypertension, white matter injury, cerebral palsy, necrotizing enterocolitis, retinopathy of prematurity, sepsis, IL-1, IL-1Ra, IL-33, IL-18, IL-37, IL-36, IL-36Ra, IL-38

## Abstract

Preterm birth is a major contributor to neonatal morbidity and mortality. Complications of prematurity such as bronchopulmonary dysplasia (BPD, affecting the lung), pulmonary hypertension associated with BPD (BPD-PH, heart), white matter injury (WMI, brain), retinopathy of prematurity (ROP, eyes), necrotizing enterocolitis (NEC, gut) and sepsis are among the major causes of long-term morbidity in infants born prematurely. Though the origins are multifactorial, inflammation and in particular the imbalance of pro- and anti-inflammatory mediators is now recognized as a key driver of the pathophysiology underlying these illnesses. Here, we review the involvement of the interleukin (IL)-1 family in perinatal inflammation and its clinical implications, with a focus on the potential of these cytokines as therapeutic targets for the development of safe and effective treatments for early life inflammatory diseases.

## 1. Introduction

Fifteen million infants are born preterm every year [1]. Defined as birth before 37 weeks of pregnancy, preterm birth is the leading cause of neonatal morbidity and mortality, with the lowest gestational age (GA) and smallest infants most at risk. Morbidities, which we will refer to as complications of prematurity, include bronchopulmonary dysplasia (BPD) and pulmonary hypertension associated with BPD (BPD-PH) affecting the lung and heart, diffuse white matter injury (WMI) of the brain, necrotizing enterocolitis (NEC) of the gut, retinopathy of prematurity (ROP) of the eyes and sepsis. Worldwide, one million infants under 5 years of age died from complications of prematurity in 2015 alone [2]. Associated healthcare costs for prematurity in the United States exceeded US$26 billion during 2005 [3], which is equivalent to $40 billion today [4]. Though the origins of preterm birth and complications thereof are heterogeneous in nature and vary between women and infants, it is understood that pre- and post-natal inflammation (here, collectively called early life inflammation) underpins the pathophysiology of most early life illnesses, particularly in the setting of prematurity.

The interleukin (IL)-1 family is a group of cytokines and receptors that facilitate inter- and intracellular communication to mediate inflammation [5]. The pro-inflammatory cytokine IL-1β has been identified as having a pathogenic role in early life diseases [6,7]; the IL-1 receptor antagonist (IL-1Ra) is the naturally occurring antagonist of IL-1β. Under the generic name anakinra, recombinant IL-1Ra has been in clinical use for over 20 years, with a well-established safety and efficacy profile for the treatment of inflammatory disease [8,9,10,11,12,13], including for infants with neonatal-onset multisystem inflammatory disease (NOMID) [14,15]. Mounting preclinical evidence suggests that IL-1Ra is protective in the complications of prematurity, including BPD, BPD-PH [16,17,18,19] and WMI [20,21,22,23]. Research into the inflammatory mediators responsible for NEC, ROP and neonatal sepsis and the role of newer IL-1 family members in all the complications of prematurity is preliminary but opens the field for further studies. This review describes the role that IL-1 family members play in perinatal inflammation and its clinical implications, with a focus on identifying key mediators to pioneer the development of new IL-1 based therapeutic strategies.

## 2. The IL-1 Family

As of today, eleven IL-1 family ligands have been described, consisting of seven pro-inflammatory cytokines (IL-1α, IL-1β, IL-18, IL-33, IL-36α, IL-36β and IL-36γ), two receptor antagonists (IL-1Ra and IL-36Ra) and two anti-inflammatory cytokines (IL-37 and IL-38). These cytokines have been organized into three sub-families based on their structural similarity with other members [5] (Table 1 and Figure 1). The family of IL-1 receptors (IL-1R) includes ten structurally related receptor molecules (called IL-1R1-10 according to the latest nomenclature [5,24]) and the distantly related soluble IL-18 binding protein (BP) that has been identified as an inhibitor of IL-18 [24]. IL-1Rs contain the cytoplasmic Toll-IL-1-receptor (TIR) domain, a structural element they have in common with Toll-like receptors (TLRs), which participate in host defense by recognizing potentially harmful molecular patterns. There are substantial similarities between IL-1 and TLR family signaling due to this shared TLR domain [25]. The notable exceptions are IL-1R2, which lacks an intracellular TIR domain [26], and IL-1R8, which comprises a TIR domain with two amino acid substitutions [27]. Extracellularly, IL-1Rs also possess three immunoglobulin (Ig)-like domains in the extracellular receptor segment, except for IL-1R8 and IL-18BP, which only contain one Ig-like domain [28]. The mechanisms of action of the IL-1 family cytokines and receptors are described in more detail below.

### 2.1. IL-1 Subfamily

The IL-1 subfamily consists of IL-1α, IL-1β, IL-1Ra and IL-33 [29]. 

#### 2.1.1. IL-1 and IL-1Ra

IL-1 was first described in 1974, originally called leukocytic pyrogen [30] before being renamed IL-1α and IL-1β 11 years later [31]. IL-1α mostly exerts its effects local to the producing cell and has been identified in many cell types, including endothelial cells [32,33]. In contrast, IL-1β is secreted predominantly by monocytes, macrophages and dendritic cells and then circulated systemically [34]. Both IL-1α and mature IL-1β are potent pro-inflammatory molecules that activate target cells via the interaction between two membrane-bound receptors, IL-1R1 and IL-1R3 [35] (Table 1 and Figure 1). Activation of the IL-1R1:IL-1R3 receptor heterodimer triggers complex signaling transduction pathways. One such pathway includes the activation of MyD88 (myeloid differentiation primary response 88), the canonical adaptor for IL-1R1 signaling (and that of most other IL-1 receptors). MyD88 activates the IRAK (IL-1R-associated kinase) family. IRAK activation leads to a variety of changes in cellular functions, which include the activation of pathways such as NF-κB (nuclear factor kappa-light-chain-enhancer of activated B cells) [36,37,38], MAPKs (mitogen activated protein kinases) [39,40,41] and AP1 (activator protein 1) [42]. Activation of these pathways induces the release of chemokines such as IL-8 [43] and other cytokines such as IL-6 [44]. The IL-1 signaling pathway is tightly regulated at multiple levels, including extracellularly by the decoy receptor IL-1R2, which lacks a cytoplasmic domain and therefore binds IL-1 (with a greater affinity for IL-1β than IL-1α) without activating a cellular response [5,45,46], and the competitive antagonist IL-1Ra, which binds IL-1R1 with high affinity, blocking the binding of IL-1 [47]. When bound by IL-1Ra, IL-1R1 does not bind IL-1R3, and therefore does not trigger the activation of downstream messengers. The drug anakinra is an analogue of endogenous IL-1Ra, with the addition of a single methionine residue at the N-terminus, which exerts the same anti-inflammatory action as IL-1Ra. Other strategies to block IL-1 [48] include the soluble IL-1 receptor, neutralizing monoclonal antibodies against IL-1β and blocking antibodies against IL-1R1. The preclinical peptide rytvela, which has been researched in the retinopathy of prematurity, also blocks the activation IL-1R1 but does so via an allosteric binding site [49].

The bioactivity of IL-1β has been found to be tightly controlled and to require protease processing (Figure 2). First, TLRs such as TLR4 are activated by DAMPs or PAMPs (danger- or pathogen-associated molecular patterns), which activate the NF-κB and MAPK pathways, amongst others, to initiate transcription and the release of pro-IL-1β into the cytoplasm [50,51]. Upon encountering a secondary signal, such as another DAMP or PAMP, the formation of the inflammasome is triggered to activate caspase-1 and cleave pro-IL-1β into its active secreted form (IL-1β) [50]. Notably, monocytes differ from macrophages in that they do not need a secondary signal to activate the inflammasome; TLR stimulation alone is enough [52].

#### 2.1.2. IL-33

IL-33 is a pleiotropic cytokine with contrasting properties in health and disease [53,54]. When secreted, IL-33 binds the membrane-bound receptor IL-1R4 and recruits IL-1R3 to activate NF-κB and MAPK and other pathways to exert its effects, notably including the differentiation of helper T cells along the type 2 pathway [53,55,56]. Under steady-state conditions, IL-33 has been shown to be produced and stored in the nucleus of barrier cells, including epithelial cells of the gut and lung. Upon cell damage, nuclear IL-33 is passively released into the extracellular space, functioning as an alarmin. Moreover, IL-33 has also been shown to be produced and released by leukocytes, including murine macrophages [57,58,59] and murine dendritic cells [59,60], after stimulation with endotoxin and pro-inflammatory cytokines, including IL-33 itself. Under healthy conditions, intracellular IL-33 participates in maintaining barrier function [54]. IL-1R4 also exists in a soluble form (sIL-1R4), acting as a decoy receptor to reduce IL-33 signaling [61]. 

### 2.2. IL-18 Subfamily

The IL-18 subfamily comprises the pro-inflammatory cytokine IL-18 and the anti-inflammatory cytokine IL-37.

#### 2.2.1. IL-18

First described in 1995 [62], IL-18 binds IL-1R5 and recruits IL-1R7 [63,64], leading to downstream activation of pro-inflammatory transcription factors, including NF-κB. Macrophages and dendritic cells have been reported to be important sources of active IL-18 [65]. IL-18 has a range of effects that are important for host defense against pathogens, which include increased cell adhesion molecules, nitric oxide synthesis, chemokine production [5,66] and interaction with IL-12 to induce interferon (IFN)-γ and type 1 polarization [67,68,69]. In a negative feedback loop, IFN-γ induces the release of the IL-18-binding protein (IL-18BP) [70], which binds IL-18 with high affinity and thereby inhibits its functions [71].

#### 2.2.2. IL-37

Interleukin 37 (IL-37) is a powerful broadly acting inhibitor of inflammation [72] that is largely uninvestigated in the setting of prematurity or complications thereof. Discovered in the early 2000s via computational cloning [73], IL-37 binds IL-1R5, but unlike IL-18, it does not recruit IL-1R7 [74]. Instead, IL-37 exerts potent anti-inflammatory signals via its receptor complex IL-1R5:IL-1R8 [75] or by shuttling to the nucleus [72]. Though the multi-faceted anti-inflammatory signal transduction program elicited by IL-37 is complex, the tripartite complex IL-37:IL-1R8:IL-1R5 assembles rapidly on the surface of PBMCs (peripheral blood mononuclear cells), for example, upon stimulation with LPS (lipopolysaccharide, a bacterial endotoxin [75]). In most PBMC subsets, IL-37 abundance is low at baseline. In contrast, monocytes and myeloid dendritic cells (mDCs) [76] store and secrete IL-37 in response to inflammatory stimuli, thus acting in an alarmin-like fashion [76]. Mice transgenic for IL-37 (IL-37tg) are protected from endotoxemia, whereas in IL-37tg mice deficient in IL-1R8, this protection is substantially weaker [75]. Moreover, IL-37 impairs the activation of IL-1β and IL-18 by inhibiting inflammasome function [77]. A small subset of mDCs constitutively express and secrete IL-37 and potentially contribute to immune homeostasis, providing an anti-inflammatory milieu at a steady state [76].

### 2.3. IL-36 Subfamily

The IL-36 subfamily comprises the pro-inflammatory cytokines (IL-36α, IL-36β, IL-36γ) and the anti-inflammatory receptor antagonist (IL-36Ra) and cytokine (IL-38).

#### 2.3.1. IL-36 and IL-36Ra

IL-36 agonists signal through a common receptor complex IL-1R6:IL-1R3 [78,79]. Similar to the mechanism of action of IL-1Ra, IL-36Ra competitively antagonizes the binding of IL-36 agonists to IL-1R6 [80]. Both the agonists and the receptor antagonist require processing to exert their activity; native IL-36α, β and γ are 100–1000 times less active than their processed counterparts, and native IL-36Ra exerts no antagonist activity [80]. Upon the receptor heterodimer IL-1R6:IL-1R3 activation by IL-36 agonists, a multitude of inflammatory pathways are activated, including those mediated by NF-κB and MAPK [78]. IL-36 agonists are capable of inducing or amplifying Th1 and Th3 immune responses in T cells and myeloid cells. Among T cells, IL-36R is predominantly expressed on naive T CD4(+) T cells; IL-36 cytokines promote T cell proliferation and IL-2 secretion [81]. IL-36β acts in synergy with IL-12 to promote Th1 polarization [81]. Unilateral ureteral obstruction (UUO) in mice deficient for IL-36R exhibit markedly reduced NLRP3 inflammasome activation and macrophage/T cell infiltration in the kidney as compared to wild-type mice subjected to UUO [82]. In vitro, recombinant IL-36α facilitated the activation of the NLRP3 inflammasome in epithelial cells, macrophages, and dendritic cells, and it also induced T cell proliferation and Th17 differentiation [82]. IL-36 dysregulation in skin leads to keratinocyte and immune cell induction of the Th17/Th23 signaling axis, and it induces a psoriasis-like skin disorder [83,84,85]. However, evidence on the physiological and pathological functions of IL-36 in other organs in which *IL36* is expressed, such as the lungs, the gut, and the brain, remain poorly studied.

#### 2.3.2. IL-38

IL-38 is a poorly characterized member of the IL-36 subfamily. Discovered 20 years ago [86,87], IL-38 has shown an affinity for IL-1R1 [87], IL-1R6 [88] and IL-1R10 [89], but it is still not clear which receptor(s) is/are essential for IL-38 signaling. IL-38 also has a strong homogeneity to the IL-1 and IL-36 receptor antagonist (39% and 43% homology in humans, respectively) [87]. IL-38 expression has been identified in many organ systems, including the skin, placenta, heart and brain [86,90]; the IL-38 protein is induced by monocytes and macrophages after stimulation with LPS in vitro [91]. IL-38 has been shown to inhibit IL-17 in multiple disease models of skin inflammation [92], liver injury [93] and arthritis [94]. IL-1R10 is required for IL-38 to suppress γδT cell IL-17 production [92]. Interestingly, the ablation of IL-38 in a mouse model of autoimmune encephalomyelitis, which is traditionally IL-17-driven, improved clinical outcomes and reduced inflammation in affected mice [95]. The role of IL-38 in perinatal inflammation has not been studied.

## 3. Preterm Birth and Sources of Fetal Inflammation

The risk of preterm birth increases with the adverse inflammatory status of the mother and infant during pregnancy [96,97]. Besides maternal ethnicity and age [98], such inflammation underpins most other known risk factors for preterm birth to a considerable degree, which include stress, drugs, smoking, alcohol, maternal autoimmune conditions (such as systemic lupus erythematosus [99], rheumatoid arthritis [100], inflammatory bowel disease [101], type 1 and 2 diabetes [102]), conditions that only manifest during pregnancy such as preeclampsia [103] and chorioamnionitis [104,105,106] and premature rupture of membranes (PROM) [107]. 

One of the most common causes of preterm birth, chorioamnionitis, results from inflammation of the chorioamniotic membranes and can precipitate an inflammatory reaction in the fetus [108]. Increased production of IL-1β in both the mother and fetus during chorioamnionitis has been reported by multiple studies, including in amniotic fluid [109,110], placenta and chorioamniotic membranes [109,111,112], the umbilical cord in the case of funisitis [113,114] and in maternal serum [115,116,117]. Chorioamnionitis often occurs following PROM [118,119], and increased production of IL-1β in chorioamnionitis is a major contributor to tight junction destruction in placental tissues [120,121]. Premature infants born to mothers with chorioamnionitis presented with elevated pulmonary IL-1β in tracheal aspirates when compared to infants that had spontaneous preterm birth without chorioamnionitis [122,123]. Systematic reviews and meta-analyses highlight that chorioamnionitis may be associated with complications in the offspring underpinned by inflammation, including sepsis [124], BPD [125], NEC [126], ROP [127] and neurological injury [128], including cerebral palsy [129].

Another complication of this inflammation, namely, fetal inflammatory response syndrome (FIRS), occurs when a fetus mounts its own inflammatory response when exposed to intrauterine infection. FIRS is characterized by elevated umbilical cord blood mediators of inflammation, including IL-6 and IL-1β, in the setting of funisitis and chorionic vasculitis (inflammation of the umbilical cord and vessels, respectively) [130,131,132]. A systematic review and meta-analysis of FIRS patients and adverse neonatal outcomes summarized that the FIRS is associated with a higher incidence of sepsis, BPD, intraventricular hemorrhage (IVH), respiratory distress syndrome (RDS) and death [133].

## 4. The Role of the IL-1 Family in Early-Life Inflammation and Injury

Though multifactorial in etiology, most complications of prematurity (Figure 3) share a common underlying pathophysiology, namely, inflammation. Of importance to note is that infants who incur one of these complications are predisposed to others. On these grounds, a better understanding of the underlying signaling pathways leading to inflammation could translate into novel targets for the prevention and treatment of the complications of prematurity.

### 4.1. IL-1 and the Preterm Cardiopulmonary System

Bronchopulmonary dysplasia (BPD) and pulmonary hypertension associated with BPD (BPD-PH) are cardiopulmonary morbidities faced by premature infants. These conditions are underpinned by a surge in pulmonary inflammation [134]. In Australia and New Zealand, 34% of infants born prior to 32 weeks’ gestation develop BPD [135]. Of these BPD infants, about 17% develop the subsequent condition BPD-PH, which carries a mortality rate as high as 14–38% [136]. Survivors present with altered lung volumes, reduced lung compliance and increased airway resistance at school age [137] and also a greater risk of elevated blood pressure [138], asthma [139] and neurodevelopmental impairment [140,141]. 

Infants born very prematurely (i.e., at less than 32 weeks’ gestation) have lungs that are in the canalicular and/or saccular phase of development [142] and have a thicker diffusion barrier, a reduced surface area for gas exchange and a deficiency in surfactant. As a result, lower gestational-aged infants are likely to develop respiratory distress and require respiratory support, such as mechanical ventilation and an increased fraction of inspired oxygen (FiO_2_), to overcome the reduced diffusing capacity of the lung [143]. Mechanical ventilation can damage the lung via multiple mechanisms due to increased transalveolar pressure, increased alveolar volume, and the repetitive collapse and re-expansion of alveoli required for air breathing [144]. This causes stress on the immature bronchial, alveolar and capillary epithelium, resulting in a local inflammatory response [134]. In addition, hyperoxia increases the production of reactive oxygen species (ROS). ROS can then react and cause cellular damage, which has functional repercussions, including the overactivation of immune cells, apoptosis, necrosis and subsequent cellular and tissue damage mediated by inflammation [145,146]. Such inflammation results in increased pro-inflammatory cytokines such as TNF (tumor necrosis factor), IL-6 and IL-1β [147,148] in the lungs of preterm infants, resulting in the arrest of alveolar and microvascular development. Fewer and larger alveoli, together with an accompanying reduction in capillary density and alveolar septal thickening, reduce the diffusing capacity of the lung [149,150]. Diminished angiogenesis decreases the cross-sectional area of the pulmonary vasculature and increases vascular resistance and ultimately pulmonary blood pressure [138,151,152,153].

Despite recognition that inflammation is the common pathogenetic pathway in the development of early life cardiopulmonary disease, there remains a lack of safe and efficacious treatments. Systemic corticosteroids are the only anti-inflammatory drugs in clinical use, but they have limited efficacy at reducing BPD/BPD-PH and are associated with serious adverse events [154], including cerebral palsy [155]. Concurrently, glucocorticoids inhibit alveolar growth, thereby impeding the process needed to recover from BPD [134,156,157].

#### 4.1.1. IL-1 Mediates Preterm Cardiopulmonary Pathophysiology

Extensive preclinical work has established IL-1 as a key driver of early-life cardiopulmonary disease. A murine “double hit” model pivotal for understanding this relationship was developed in 2013 by subjecting pregnant mice to an intraperitoneal (i.p.) injection of 150 µg/kg lipopolysaccharide (LPS) at day 14 of gestation (hit 1) and subsequently placing the pups in hyperoxia (65% O_2_) to develop BPD/BPD-PH (hit 2). Pups reared in room air (21% O_2_) acted as control animals [16,158]. The manifestation of murine BPD and BPD-PH in this model is precipitated by a rise in pulmonary inflammation. In the lungs of 3-day old pups, antenatal inflammation alone elevated IL-1α 2.5-fold, but when combined with hyperoxia, IL-1β was increased 20-fold [16]. The placement of pups in hyperoxia (65% O_2_) for 28 days (end of alveolar stage of lung development in mice) resulted in a change in the pulmonary architecture, with emphysematous changes to lung morphology, including reduced alveolar number (24–44%), increased alveolar size (40–230%) and reduced surface:volume ratio (18–35%), all of which were exacerbated with increased oxygen exposure [16,17,159]. In addition, the double-hit mice exhibited increased airway smooth muscle proliferation and airway hypercontraction to the muscarinic acetylcholine receptor agonist, methacholine (MCh) [17]. Furthermore, the number of small blood vessels (4–5 μm diameter) was reduced by 84% as measured by micro-CT, and pulmonary vascular resistance was increased as measured by echocardiography [18,19]. Daily administration of anakinra (10 mg/kg) to neonatal mice abolished the increases in IL-1α and IL-1β at day 3 of the model and prevented the structural damage to the alveoli seen at 28 days [16]. Anakinra also afforded protection against increased airway smooth muscle, reduced the number of pulmonary small vessels and increased pulmonary vascular resistance in vivo, but did not counteract the airway hyperreactivity to MCh [17,18,19]. 

The IL-1-driven pathology in early life has been confirmed in other in vivo models of antenatal inflammation and BPD, including in rodents, sheep and baboons. Intra-amniotic injection of 100 μg IL-1α to sheep 1, 3 or 7 days prior to delivery at 124 days’ gestation (equivalent to 34 weeks of human GA) was accompanied by increased immune cell recruitment in the bronchoalveolar lavage fluid from fetal lungs (neutrophils, monocytes and lymphocytes) [160]. Neonatal wild-type (WT) mouse pups reared in 85% O_2_ from postnatal days (P)3-14 incurred a 40-fold increase in whole lung *Il1b* mRNA on P10 and a 10-fold increase in bronchoalveolar lavage (BAL) IL-1β protein on P15. Pups deficient in NLRP-3, one of the inflammasomes responsible for the cleavage of IL-1β into its active form, reported a similar increase in *Il1b* mRNA, but not IL-1β protein. Notably, WT pups had a simplified distal lung architecture and larger saccular structures, and NLRP-3 knockout (KO) pups had a normal lung architecture and well-formed alveoli, highlighting the importance of the NLRP-3 inflammasome and IL-1-mediated inflammation in the pathophysiology of BPD [161]. A primate study investigated the effect of ventilation and oxygen exposure on lung inflammation in preterm baboons. Preterm baboons were delivered via cesarean section at different gestations, and tracheal IL-1β was measured. A gradual increase in the IL-1β:IL-1Ra ratio from 125 days to 185 days’ gestation (equivalent to a human infant at 28 weeks’ gestation to term) was observed, which was in keeping with the increasing inflammatory state [161]. Premature baboons, delivered at 125 days’ gestation, treated with exogenous surfactant and ventilated for 14 days presented with *IL1B* mRNA expression up to 60-fold higher, IL-1β protein 3-fold higher and interrupted alveolarization, as compared to the non-ventilated fetal baboons harvested at 125 and 140 days’ gestation. Accordingly, the IL-1β:IL1Ra ratio was increased in the ventilated baboons, further highlighting the immune dysregulation in BPD [162,163]. These examples highlight the mounting evidence in pre-clinical models of the pathological role of IL-1 in development of BPD.

Though there is ample preclinical evidence of the role of the IL-1 subfamily in damage to the respiratory system in early life, the role of newer IL-1 family members is less clear cut. IL-33 production has been observed to be initiated from the pressures exerted by the first breath of neonatal mice, inducing a type-2 immune environment [164]. A high IL-33 environment during the alveolar phase of lung development has been shown to lead to a Th2 response with infiltration of eosinophils, mast cells and basophils, resulting in an increased incidence of airway hyperreactivity [165]. This suggests that when the immature lungs of premature infants are exposed to air breathing, with or without mechanical ventilation, resultant IL-33 production could lead to the development of asthma. In a murine BPD model, IL-33 has also been observed to be increased on P3 compared with non-BPD mice, believed to be induced by hyperoxia [19]. A transgenic mouse model induced to overexpress IL-33 in the lung epithelial cells of neonatal mice resulted in a mortality of 61% by P14, with affected mice exhibiting a simplified alveolar structure and increased size consistent with BPD pathology [166]. Interestingly, overexpression of IL-33 in adult mice, or neonatal IL-1R4-KO mice, did not have increased mortality or emphysema [166]. Another murine model induced BPD by the intra-amniotic injection of LPS (1 μg) to pregnant mice on day 16.5 of gestation, and pups administered with 0.2 μg recombinant mouse IL-33 i.p. on P7 exhibited aggravated BPD changes. Conversely, in pups administered with 0.5 μg anti-IL-1R4 antibody i.p. every alternate day from P7, lung architecture was preserved [167]. Moreover, blocking IL-33 and/or its receptor improved disease outcome in other murine models of BPD [168,169]. Collectively, these studies highlight the pathogenic role of IL-33 in the premature lung in which the timing of IL-33 exposure is important, and they demonstrate that blocking IL-33 signaling might prove useful for the prevention of BPD. 

Another pro-inflammatory IL-1 cytokine, IL-18, was elevated in the lungs of neonatal rodents subjected to hyperoxia or environmental pollutants [170,171]. This IL-18-mediated pathology warrants further investigation. Recently, the anti-inflammatory IL-37 has been suggested to be a potential therapeutic agent in inflammatory disease [172]. Though largely undescribed in early life, IL-37 has been shown to be beneficial in an animal model of neonatal respiratory distress syndrome (RDS) [173]; pre-treatment with 1 μg of recombinant human IL-37 ameliorated pathological changes [173] induced at P6 old mice injected i.p. with 10 mg/kg LPS. Pups presented with elevated pulmonary inflammation (IL-1β, TNF, IL-8 and CCL2), apoptosis and lung pathology (alveolar congestion, hemorrhage, edema and inflammatory cell infiltration) within 24 h of LPS plus vehicle injection, whereas IL-37-treated pups were protected [173]. Though there is strong evidence for the role of the IL-1 subfamily members in the development of BPD, more research is needed on the role of IL-36 and IL-18 subfamily members.

#### 4.1.2. Clinical Association between IL-1 and Preterm Cardiopulmonary Morbidity

In the literature, IL-1β is the most prevalent cytokine in predicting infants at risk of cardiopulmonary morbidity. In 25 infants with respiratory distress syndrome, the 12 infants whose initial bronchoalveolar lavage (BAL) sample (within first hour of life) was positive for IL-1β were associated with chorioamnionitis (clinical and histologic) and were smaller in size, more immature and required a longer time on mechanical ventilation and supplemental oxygen therapy than the 13 infants whose initial BAL sample was negative for IL-1β [122]. Both serum and BAL IL-1β levels were significantly higher in infants with BPD compared to non-BPD infants [174]. Serum from premature infants taken on day 14 of life and analyzed for inflammation revealed that infants ventilated for two weeks had an elevated level of IL-1β compared to the infants ventilated for less than 7 days [175]. Serial BAL samples taken from 16 infants with BPD revealed an elevated IL-1Ra on the first day of life compared to 19 infants without BPD. This early abundance of IL-1Ra, though initially protective, proves suboptimal due to the stronger abundance of IL-1β as seen by day 5 and 7 [176]. Intubated premature infants had increased mRNA expression of *IL1A*, *IL1B* and *IL1RN* (gene for IL-1Ra) by day 7 of life in tracheal aspirates of infants that went on to develop BPD [177]. Mesenchymal stromal cells taken from tracheal aspirates of premature infants with severe BPD exhibited an increase in NF-κBp65 (a key transcription factor of the IL-1-signaling pathway). In vitro stimulation of mesenchymal stem cells with IL-1β confirmed an increase in NF-κBp65 [178]. 

Another emerging biomarker to diagnose, monitor and guide the treatment of early life pulmonary disease is IL-33. Serum IL-33 was elevated at the time of BPD diagnosis at 36 weeks corrected GA in infants born prematurely [179]. Moreover, there was elevated serum IL-33 on days 1, 14 and 28 of life in infants that developed BPD compared to those that did not [180]. IL-33 consistently decreased after hydrocortisone treatment in infants with BPD, and therefore, serum IL-33 can be used to monitor an infant’s response to treatment [179,180]. Another study found conflicting results, that IL-33 in cord blood and peripheral blood from preterm infants on day 14 post birth was not associated with BPD. Instead, this study detected that soluble IL-1R4 was closely associated with BPD severity [181].

The pro-inflammatory IL-18 was also found to be elevated postnatally on day 14 in the serum of infants that developed BPD [182]. However, polymorphisms in the gene coding for IL-18 did not translate to increased incidence of BPD [183], but single nucleotide polymorphisms (SNPs) of the IL-18 receptors (IL-1R5 and IL-1R7) were associated with BPD in an African American (AA) (but not Caucasian) population [184]. In summary, there is a strong link between aberrantly elevated IL-1β and its pathogenic role in BPD, though the evidence for other IL-1 family members is sparse.

### 4.2. IL-1 and the Preterm Brain

Extremely premature infants are at high risk of long-term neurodevelopmental complications, including cerebral palsy, intellectual disability, microcephaly and autism spectrum disorders [185], which all are inversely correlated with GA and birth weight. Injury to the developing white matter is one of the most common pathological substrates seen in preterm infants. White matter injury (WMI) can be diffuse or focal and result in cystic (necrotic) or non-cystic lesions that lead to reduced axonal myelination within the affected areas [186]. Premature infants are also at increased risk for intracranial/intraventricular hemorrhage (ICH/IVH), which in itself invokes an inflammatory response in surrounding tissue [187]. Magnetic resonance imaging (MRI) performed at term-corrected age in infants born between 23–30 weeks’ gestation demonstrated diffuse WMI in up to 80% of the cohort and subsequent neurodevelopmental impairment in up to 35% [141,188]. The cause of WMI is multifactorial, with risk factors including a multitude of pre- and postnatal factors (e.g., chorioamnionitis, hypoxia/ischemia (HI), ICH/IVH and postnatal sepsis), which trigger systemic and central nervous system inflammation [189,190,191,192,193,194,195].

The term “white matter” refers to the paler tissue of the brain and spinal cord, comprising primarily nerve axons and myelin. Myelin is a multi-layered glial membrane that surrounds axons to provide nutrition and to insulate axons, thus increasing the conduction speed of action potentials [196]. The production of myelin occurs predominantly after 32 weeks’ gestation by mature oligodendrocytes [186,196]. Between 23 and 30 weeks, the oligodendrocyte precursors, pre-myelinating oligodendrocytes, make up the majority of oligodendrocyte lineage. These immature oligodendrocytes are particularly vulnerable to inflammation; exposure to excessive inflammation causes WMI via either apoptosis of, or impaired maturation of, immature oligodendrocytes, which results in reduced numbers of mature oligodendrocytes and reduced myelination [186,194,196].

There are currently no anti-inflammatory therapies for the prevention of WMI in preterm infants. In fact, the only commonly used treatment targeting inflammation, namely, glucocorticoids, can exacerbate brain injury and increase the risk of cerebral palsy [197]. The potential for glucocorticoids to cause deleterious effects in the preterm brain relates to the stage of neurodevelopment at the time of exposure, and the dose and duration of exposure relative to the insult, as previously reviewed [198,199], though the underlying mechanism remains unknown. Magnesium sulfate (MgSO_4_) for preterm neuroprotection, currently recommended for maternal administration when preterm labor is expected before 30 weeks’ gestation, may in part act through inhibition of the NF-κB inflammatory pathway [7,141,194]; however, recent follow-up studies to school age suggest it does not significantly improve longer-term neurodevelopmental outcomes compared to placebo [200,201]. Furthermore, preclinical evidence has shown that MgSO_4_ may be associated with the loss of oligodendrocytes, likely due to the NMDA-induced inhibition of oligodendrocyte development [202]. MgSO_4_ has also been associated with a lower risk of type 2 immune polarization in the infant, reducing the risk for BPD, though the significance of this to neurodevelopment is unknown [19]. Collectively, these data suggest that current therapies aimed at improving neurodevelopmental outcomes of premature infants are variably effective, and the development of targeted anti-inflammatory treatments is urgently needed.

#### 4.2.1. IL-1 Family-Mediated Pathophysiology of Preterm Brain Injury

IL-1β is a key mediator in the pathogenesis of neonatal brain injury. As in the lung, early life animal models have been essential in investigating the role of inflammation in early life brain injury. The effects of HI injury following birth asphyxia have been widely investigated in both term and preterm animal models (mouse, baboon, sheep), with a consensus that excessive IL-1β exacerbates WMI in the fetus [6,194,203]. Neonatal rats administered with 10 ng of recombinant IL-1β via intracerebral injection into the left hemisphere on day 5 after birth exhibited increased astrogliosis and apoptotic cell death and developed oligodendrocyte loss 24 h after injection [204]. Newborn mice administered with 10 µg/kg recombinant mouse IL-1β twice daily via i.p. injection over 5 days presented with impaired oligodendrocyte maturation and long-lasting myelination defects at P30 [205]. Therefore, localized and systemic injections of IL-1β are sufficient to induce WMI in the neonatal rodent brain. 

In a sheep model of LPS-induced fetal inflammation, repeated i.v. injections of IL-1Ra reduced circulating cytokines (IL-1β, TNF, IL-6 and IL-10) and improved the recovery of fetal movement and electroencephalogram activity compared to LPS plus vehicle [20]. In line with the improvements in neurophysiology, histopathological analyses revealed less microgliosis, reduced accumulation of IL-1β and improved pre-oligodendrocyte survival in the large white matter tracts of the LPS+IL-1Ra group when compared to LPS plus vehicle, demonstrating that IL-1Ra protected against inflammation-induced WMI [20]. Similarly, neonatal rodent studies have shown that IL-1Ra-induced reductions in placental and CNS inflammation and improvements in oligodendrocyte survival, myelination and neurobehavioral outcomes after exposure to inflammatory and or hypoxic insults [21,22,23]. Though an overabundance of IL-1 adversely affects neurodevelopment, so too does too little, with animal studies using knockout or transgenic models to obliterate IL-1 signaling showing adverse effects to hippocampal volume, memory and behavior [159,206,207]. These studies highlight the importance of the balance of the IL-1:IL-1Ra ratio for normal neurodevelopment [159].

Neonatal rodent studies suggest a neuroprotective role for IL-33 in some experimental settings. For example, IL-33 promoted the release of neurotrophic factors from astrocytes essential for neuronal survival against oxygen-glucose deprivation [208]. Neonatal mice administered with recombinant mouse IL-33 for 3 days after HI injury on P7 incurred less brain damage as compared to mice injected with saline [208]. Conversely, IL-1R4 (IL-33 receptor) deficiency exacerbated brain infarction and neurological injury after HI insult [208]. Recurrent neonatal seizures (RNS) can be modeled in rats via the inhalation of volatile flurothyl from P7 for 7 days. RNS mice develop a phenotype of neurobehavioral deficits, weight loss and apoptosis. Prophylactic i.p. administration of recombinant mouse IL-33 (300 ng) prevented RNS [209,210]. The role of IL-18 has also been investigated in rodent studies. In P9 neonatal mice, knockdown of IL-18 improved myelination and axonal integrity after HI brain injury [211]. Indeed, reduced IL-18 has been associated with diminished WMI in several rodent models of neonatal brain injury [212,213,214,215,216,217,218]. However, further preclinical studies are required to understand the mechanisms that underpin the potential protective roles of IL-33 and reduced IL-18 in the preterm brain.

#### 4.2.2. Clinical Association between IL-1 and Preterm Brain Injury

There is a strong association between excessive IL-1β and preterm brain injury. A systematic review from 2010 on cytokine abundance in maternal, cord and postnatal blood highlights that elevated IL-1β is associated with neurologic damage [219]. In both extremely premature and late preterm infants, elevated systemic and cerebrospinal IL-1β during the first 48 h of life were associated with impaired cerebral metabolism and developmental delay at 2 years of age. Moreover, in postmortem preterm brain tissue, areas of WMI had increased immunoreactivity of IL-1β, IL-1R1 and IL-1R2 on astrocytes and microglia [220]. In the same study, IL-1Ra abundance was also increased, but to a lesser extent than IL-1β, suggesting an imbalance towards pro-inflammatory mediators at sites of tissue damage [220]. In neonatal sepsis, plasma, serum and cerebrospinal fluid (CSF) IL-1β were elevated in infants at a greater risk of neurological impairment [221,222,223]. In preterm infants with damaged periventricular white matter, known as periventricular leukomalacia (PVL), who later developed cerebral palsy, cord blood IL-18 was higher compared to healthy term infants [224]. 

Collectively, this growing body of clinical evidence and preclinical mechanistic studies suggests that targeting IL-1β could be an effective therapeutic approach for promoting neuroprotection in the preterm brain. Other IL-1 family cytokines could also be targets of interest, though the evidence is less advanced.

### 4.3. Other Complications of Prematurity

#### 4.3.1. Retinopathy of Prematurity

Retinopathy of prematurity (ROP) is the most common cause of blindness in premature infants. ROP results from inhibition of the growth of retinal vessels followed by abnormal proliferation, which in the most severe cases can result in retinal detachment [49,225,226]. Hyperoxia is an important driver of retinal inflammation, activating microglia to produce pro-inflammatory cytokines such as IL-1β, resulting in arrested vascular growth; this leads to hypoxia of the distal retina, which triggers abnormal vasculogenesis [49,227,228,229]. Blockade of IL-1β using anakinra, as well as a selective allosteric inhibitor of the IL-1 receptor, rytvela, have been trialed in animal models of ROP [230,231].

The inflammatory pathophysiology of ROP begins with antenatal exposure to inflammation [231]. To compare the effect of IL-1β inhibition by anakinra and rytvela antenatally, a mouse model of antenatal inflammation was used. Dams were injected subcutaneously with anakinra 4 mg/kg, rytvela 1mg/kg or placebo at 12h intervals from gestational day 16.5 through 18; 30 min following the initial injection, a single intrauterine injection of 1 ug of IL-1β was administered. Three to five pups per group were sacrificed at days 17, 17.5 or 18, and their eyes were examined. Pups born at full gestation of 19 days had regular eye imaging over the first 3 weeks; others were sacrificed at P1, 4, 8, 15, 22 or 30 for histological eye examination. Eyes of pups within the placebo group exhibited a higher concentration of activated microglia and associated damage when compared to pups from dams administered with anakinra or rytvela [231].

Dual hits, such as postnatal exposure to hyperoxia, intensified antenatal inflammatory responses mediated by TNF and IL-1β [232]. Pups exposed to 80% oxygen for 20 hr at P6 revealed a four-fold increase in retinal *Il1b* mRNA compared to rats raised in normoxia, with no increase in endogenous IL-1Ra until P10. These eyes exhibited an influx of activated microglia and retinal vaso-obliteration consistent with ROP, which was prevented by treatment with anakinra [229]. An oxygen-induced retinopathy (OIR) rat model has been established, whereby rats born at term were exposed to oxygen levels that cycled between 50% and 10% every 24 h from P0 to P14, then remained at 21% thereafter; control pups were reared only in 21% oxygen [49,232]. At P14, pups reared in hyperoxia had higher IL-1β in retinal tissues, choroidal thinning and subretinal hypoxia (measured by hypoxyprobe staining) compared with those raised in normoxia [232]. With the application of the same OIR model, pups received i.p. injections of anakinra 20 mg/kg, rytvela 1 mg/kg or no treatment to serve as controls, and their eyes were imaged at P14 and P30, before being sacrificed at P30 to examine eye histology. At P30, retinal thinning was observed in the untreated OIR group compared with the treated and normoxia groups on retinal imaging [49]. Histology of the same eyes showed a reduction in preretinal neovascularization, suggesting IL-1R1 blockade is protective against ROP [49,232].

Human studies examining vitreous fluid and tear samples from infants with and without ROP illustrate such inflammatory changes. Vitreous fluid taken from infants with severe ROP at the time of laser treatment had more activated microglia and macrophages, along with significantly increased cytokine abundance, including of IL-1Ra, when compared to control samples taken from infants with congenital cataracts [233]. Interestingly, tears of infants with severe ROP had significantly higher abundance of the anti-inflammatory IL-1Ra compared with those with mild or no ROP, pointing to a counter-regulatory mechanism [233]. 

The role of IL-33 in the pathology of ROP remains unknown; however, increased IL-33 has been associated with severe ROP, suggesting that IL-33 could be used as a biomarker for ROP. In an observational study on infants born less than 32 weeks’ gestation with a birth weight less than 1500 g, IL-33 was evaluated in cord blood and serum from ROP infants pre- and post-laser treatment and compared to gestation- and weight-matched controls. Cord blood IL-33 was similar between control and ROP groups, although pretreatment levels in infants with ROP rose to 3.5 times those of controls, suggesting that IL-33 could be used as a biomarker for ROP [225]. Similarly, the role of IL-37 in ROP is also unclear, with only one study investigating the use of recombinant IL-37 in a mouse ROP model [234]. Mice reared in 75% oxygen from P7 to P12 and then returned to normoxia were injected i.p. with placebo or IL-37 at 1, 5 or 20 ng/gram body weight on P12, P14 and P16. Eyes examined on P17 revealed those treated with IL-37 had increased neovascularization, in a dose-dependent manner, suggesting IL-37 promotes pathological angiogenesis. In a sub-group co-treated with anti-IL-37, this effect was inhibited [234].

#### 4.3.2. Necrotizing Enterocolitis

Necrotizing enterocolitis (NEC) is the most common intestinal disease in preterm infants and is a major cause of morbidity and mortality, particularly in infants born prior to 32 weeks’ gestation or weighing less than 1500 g [235,236,237]. In infants, the mortality rates of NEC average around 20–30% and can be as high as 70% in infants that require surgical treatment [238,239,240]. The pathogenesis of NEC is poorly understood, but several risk factors are known to contribute. Prematurity, intestinal microbial imbalance (i.e., dysbiosis), genetics, formula feeding and impaired intestinal barrier function increase the likelihood of infants developing intestinal microbial imbalance and dysregulated immune responses that lead to severe intestinal inflammation, pneumatosis and tissue necrosis characteristic of NEC [240,241,242,243]. Mimicking a multifactorial disease, such as NEC in vivo, NEC can be established by subjecting animals to one or a combination of the following stress factors: hypoxia, asphyxia formula feeding, cold stress, LPS and/or enteric bacteria derived from an infant with NEC [244,245]. Altered abundance of several IL-1 family cytokines in gut tissues and in blood has been linked to NEC.

Intestinal tissue resected from NEC infants revealed elevated pro-inflammatory (*IL1B*, *IL1A*, *IL36A*, *IL36B*, *IL36G*) and reduced anti-inflammatory (*IL37*, *IL1R8*) mRNA expression of IL-1 family cytokines compared to tissue resected from the same infants at re-anastomosis following recovery from NEC and non-inflamed tissue from healthy control infants [246]. Serum IL-1β [247], IL-18 [248] and IL-33 [249] were increased in NEC infants. IL-1β and IL-18 also increased in intestinal tissue from rats and mice subjected to experimentally induced NEC [250,251,252]. IL-1Ra was significantly reduced in buccal cells of NEC infants compared to age-matched healthy controls [253]. Blockade of IL-1β with anakinra in a neonatal rat model of NEC resulted in a reduction in IL-1β intestinal tissue levels and overall tissue injury scores [254]. Formula supplementation of the probiotic Lactobacillus rhamnosus GG was protective in a mouse NEC model and was associated with reduced IL-1β and increased IL-1R8 [252]. A similar effect was observed in rats; the addition of Bifidobacterium adolescentis to formula significantly reduced histological injury scores and increased *Il1r8* expression in intestinal tissue 3-fold [255].

A number of *IL1R8* gene mutations, resulting in non-functional variants, have been identified in infants that have succumbed to NEC [256]. Transgenic mice that were created via CRISPR-Cas9 technology with a premature stop codon, resulting in the *Il1r8* TIR domain being truncated at amino acid 174, exhibited spontaneous small intestine inflammation [257]. Although dam-fed, IL-1R8-KO pups presented with spontaneous intestinal inflammation. Induction of NEC increased IL-1β in IL-1R8-KO pups when compared to controls [258]. Altogether, the findings indicate a critical role of IL-1R8 in controlling intestinal inflammation.

In addition, anti-inflammatory immune cell subsets, such as systemic IL-37+CD45+ leukocytes, were lower at birth in preterm infants compared to term infants. The percentage of IL-37+CD45+ leukocytes is decreased in infants that develop NEC at weeks 1–2 of life (time period when NEC most commonly develops) [259]. 

In a neonatal mouse model of NEC, newborn mice were separated from their dams, fed formula and subjected to brief periods of asphyxia and 4 °C cold stress for 72 h [259]. WT mice presented with severe intestinal inflammation, pneumatosis and tissue necrosis; however, IL-37tg mice were protected from intestinal injury and NEC-associated mortality [259]. These data demonstrate that IL-37 holds substantial promise as a new therapeutic in this field.

#### 4.3.3. Sepsis

Neonatal sepsis refers to a systemic infection of the newborn and is the cause of substantial morbidity and mortality. Neonatal sepsis is categorized as either early-onset, usually defined as occurring within the first 3 days after birth [260], or late-onset, which occurs after day 3. In the majority of EOS cases, the source of the infecting pathogen can be attributed to in utero infection or to maternal flora during the birth process. The cause of late-onset sepsis is mostly acquired from the environment, for example, from indwelling cannulae. The accurate early diagnosis of neonatal sepsis remains difficult, as signs and symptoms are non-specific, and widely used blood markers such as C-reactive protein (CRP) are also non-specific and evolve over time [261]. As such, finding an early, accurate and consistent blood marker for neonatal sepsis is the subject of much research. Though unwell infants are often treated for suspected sepsis, an infecting organism is not always identified. Preterm infants are particularly at risk in the postnatal period due to their immature immune system and requirement for hospitalization [262,263].

During neonatal sepsis, as a response to a pathogen, a wide range of systemic inflammatory markers increase in the infant. Given the causal role of inflammation as a driver of adverse neonatal outcomes, sepsis has now been recognized as a major risk factor for BPD [264,265,266], BPD-PH [267] and WMI [219,221,268], among other complications. 

A strong clinical association between IL-1β-mediated inflammation and neonatal sepsis has been widely reported in the literature. Reports from as early as 1993 revealed increased IL-1β abundance in plasma from 10 newborns with clinical sepsis, in addition to increased TNF and IL-6, as compared to 22 healthy controls [269]. Since then, many clinical trials have informed on the pathophysiological role of IL-1β in neonatal sepsis, including in umbilical cord blood from infants with early onset sepsis [270,271], or increased plasma IL-1β [269,272,273,274] and serum IL-1β in septic infants [275,276]. However, selected studies differ and do not confirm increased IL-1 [277,278,279,280]. Another study considered IL-1β inferior at predicting sepsis when compared to CRP, TNF and IL-6 [273]. However, increased IL-1α in plasma or serum of infants with either late-onset sepsis [281], neonatal sepsis or meningitis shows promise as a predictive biomarker for neonatal sepsis [282].

In a prospective multicenter study, plasma from 21 infants with clinical sepsis was analyzed for biomarkers, including IL-1Ra, IL-6, circulating intercellular adhesion molecule-1 (clCAM-1) and CRP, in a ten-day period over a septic episode and compared to 20 infants with no infection [283]. IL-1Ra and IL-6 increased significantly 2 days before the diagnosis of sepsis, and both emerged superior at predicting sepsis when compared to clCAM-1 and CRP one or more days before clinical diagnosis [283]. 

Sepsis can be modeled in neonatal mice with an i.p. injection of adult cecal slurry (consisting of cecal contents suspended in dextrose, to simulate bowel perforation and peritonitis) [284]. Single prophylactic administration of IL-1Ra (100 μg/mouse i.p.) prior to slurry injection in neonatal WT mice did not increase survival [285]. Moreover, the blockade of IL-1 signaling with genetic (IL-1β-KO and caspase-1/11-KO) and pharmacological targeting (prophylactic treatment with 100 μg/mouse anti-IL-1β i.p.) of IL-1β did not increase survival either. Conversely, IL-1α-KO mice had increased survival as compared to WT mice, but when prophylactic treatment of 100 μg/mouse anti-IL-1α was administered i.p. to WT mice, only a trend for reduced mortality was observed. Collectively, the results of this study suggest that IL-1α, but not IL-1β, is responsible for IL-1R1-dependent neonatal murine sepsis lethality, and that pharmacological inhibition of IL-1α has little therapeutic value, likely due to the difficulty of neutralizing the local production and paracrine action of IL-1α with a systemic treatment [285].

Amongst other IL-1 family members, IL-33 emerges as a potential biomarker for the prediction of neonatal sepsis. In a cohort of 152 neonates at risk of early-onset sepsis, serum IL-33 was an independent predictor of sepsis, with greater predictive power when combined with progranulin and procalcitonin. An earlier study also reported an increased abundance of serum IL-33 upon the diagnosis of sepsis, which decreased on the 3rd and 7th day of antibiotic treatment [286].

Research into other IL-1 family members is more preliminary. IL-18-KO neonatal mice were highly protected from polymicrobial infection [287], and blocking IL-17A reduced IL-18-potentiated mortality to neonatal sepsis and endotoxemia [287]. In summary, the IL-1 subfamily has been associated with excessive inflammation in neonatal sepsis. Interest remains in finding a sensitive and specific marker to aid in the prompt diagnosis of the septic neonate.

## 5. Conclusions

Premature birth is a significant cause of neonatal morbidity and mortality. There is a wealth of evidence that the underlying pathophysiology of the complications of prematurity is inflammation, which is often initiated in utero by maternal inflammation and perpetuated by the challenges of post-natal life. This review highlights the role that IL-1 family cytokines play in the development of the complications of prematurity and identifies the areas in which further research is needed. The best-known member of the IL-1 family, IL-1β, plays a crucial role in the inflammatory damage to the lung, brain, eye and gut in premature infants, and preclinical models demonstrate using IL-1Ra in this population can curb the inflammatory cascade and reduce injury. Therefore, IL-1Ra shows promise as a therapeutic agent to reduce the risk of infants born extremely prematurely developing complications such as chronic lung disease, cerebral palsy, NEC and retinopathy of prematurity. Research into newer IL-1 members is less advanced. Preclinical animal models of complications of prematurity have identified a pathogenic role of IL-18 in lung and brain tissue and IL-33 in lung tissue. Moreover, a potentially protective role for IL-33 in the brain and IL-37 in the lung and gut has been identified, but these findings should be confirmed with further studies. In humans, some of these cytokines may assist the prediction of conditions of prematurity, including IL-33 for BPD and IL-18 for both BPD and PVL. IL-36 subfamily cytokines remain understudied in relation to the complications of prematurity but have been associated with NEC. Further research into the IL-1 family of cytokines could uncover a range of novel biomarkers and therapeutic targets for the better prediction and prevention of neonatal disease.

## 6. Patents

The Hudson Institute at Monash University (M.F.N. and C.A.N.-P.) holds two patent families on IL-37, namely, PCT/AU2016/050495 and PCT/EP2020/087031, and one patent on IL-38 PCT/AU2022/05146.

## Figures and Tables

**Figure 1 ijms-24-02795-f001:**
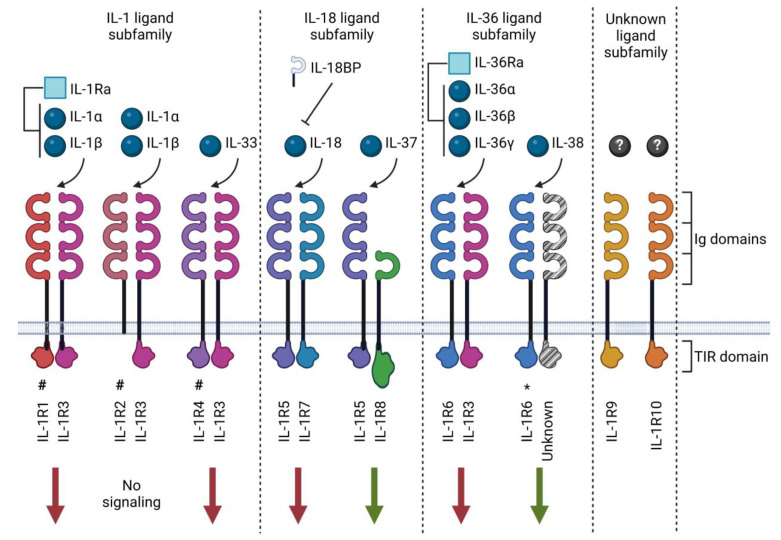
IL-1 family ligands and receptors. The IL-1 family of cytokines comprises ligands (spheres) and receptor antagonists (squares). This cytokine family utilizes ten receptors and one structurally related binding protein known as IL-18BP. The receptors may exert pro-inflammatory (red arrows) or anti-inflammatory (green arrows) signals. Most receptors contain three immunoglobulin (Ig) domains and a Toll-IL-1-receptor (TIR) domain, excluding IL-1R8, which only has one Ig domain. IL-1R2 lacks an intracellular TIR domain, and IL-1R8 has a TIR domain with two amino acid substitutions. Receptors act as scavengers in soluble form (#); receptors for which ligand binding is preliminary (*); orphan receptors with no well-established ligands (?). Created with biorender.com.

**Figure 2 ijms-24-02795-f002:**
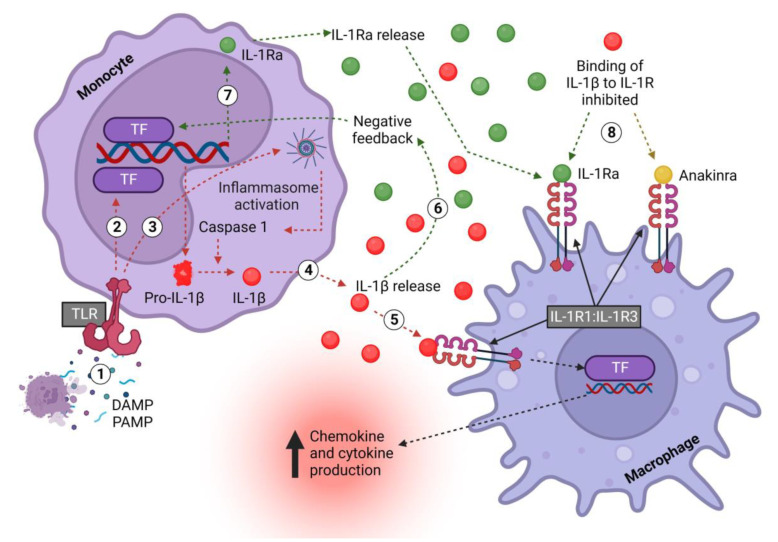
Monocyte-derived IL-1β and IL-1Ra signaling pathway in macrophages. (1) DAMPS/PAMPS bind TLRs on monocytes and (2) activate downstream TFs, which induce the transcription of pro-IL-1β and release into the cell cytoplasm. (3) In monocytes, TLR signaling also induces the formation of inflammasomes, which activate caspase-1, which in turn cleaves pro-IL-1β, thus converting it into the mature and biologically active form. (4) Active IL-1β is released to (5) activate the IL-1R1:IL-1R3 receptor complex, thus exerting multiple, mostly pro-inflammatory effects, including inducing the transcription of chemokines to promote immune cell recruitment and cytokines to promote inflammation. (6) Concurrently, pro-inflammatory mediators, including IL-1β itself, (7) increase the production of IL-1Ra. (8) IL-1Ra is released and competitively inhibits the binding of IL-1 to IL-1R1. The drug anakinra acts in an identical fashion. IL—interleukin; Ra—receptor antagonist; DAMP—damage-associated molecular pattern; PAMP—pattern-associated molecular pattern; TLR—Toll-like receptor; R1—receptor 1; TF—transcription factor. Created with biorender.com.

**Figure 3 ijms-24-02795-f003:**
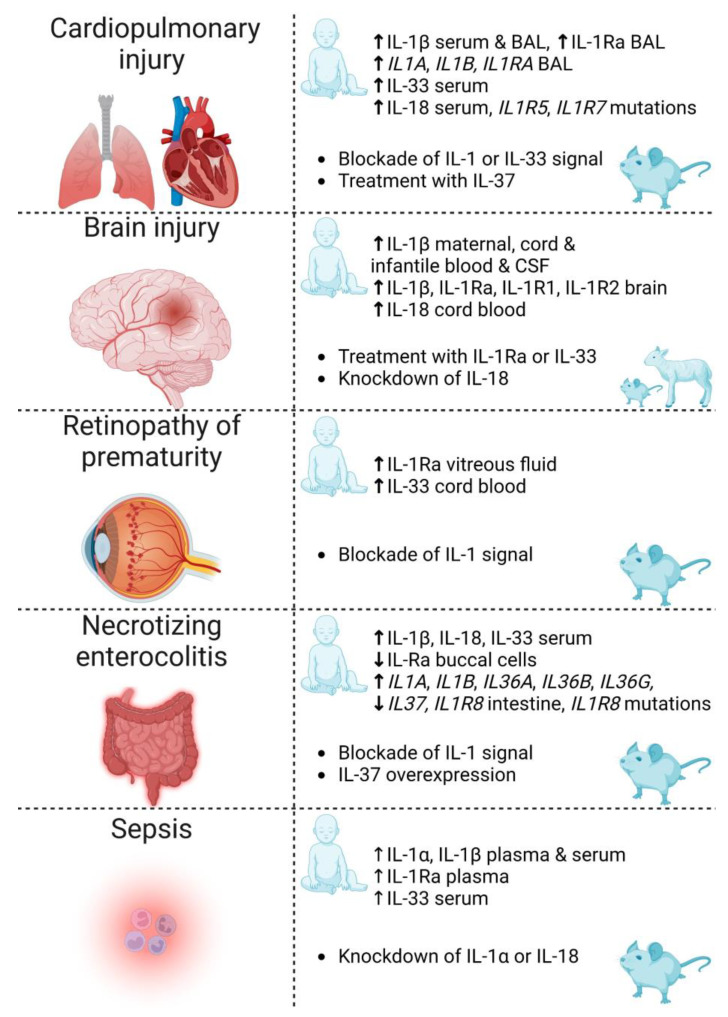
Summary of IL-1 family cytokines and receptors discussed in this review for which there is evidence of involvement in the complications of prematurity: cardiopulmonary injury, brain injury, retinopathy of prematurity, necrotizing enterocolitis and sepsis. Reported evidence includes clinical associations (human infant) and therapeutic strategies investigated in animals (rodents and sheep). IL—interleukin; BAL—bronchoalveolar lavage; Ra—receptor antagonist; CSF—cerebrospinal fluid. Created with biorender.com.

**Table 1 ijms-24-02795-t001:** IL-1 family ligands and receptors.

Sub- Family	Cytokine Ligand Name				Primary Receptor	Co-Receptor	Predominant Function Pro-/Anti- Inflammation	Receptor Antagonists and Decoy Receptors		
IL-1	IL-1α (IL-1F1) P01583		IL-1β (IL-1F2) P01584		IL-1R1 (CD121a) P14778	IL-1R3 (IL-1RAcP) Q9NPH3	Pro	IL-1Ra (IL-1F3) P18510	sIL-1R1	IL-1R2 sIL-1R2 (CD121b) P14778
	IL-33 (IL-1F11) O95760				IL-1R4 (T1/ST2, IL-1RL1, IL-33R) Q01638	IL-1R3 (IL-1RAcP) Q9NPH3	Pro	sIL-1R4		
IL-18	IL-18 (IL-1F4, IGIF) Q14116				IL-1R5 (IL-18Rα, IL-1Rrp, CD218a) Q13478	IL-1R7 (IL-18Rβ, IL-18RacP, CD218b) O95256	Pro	IL-18BP O95998		
	IL-37 (IL-1F7) Q9NZH6				IL-1R5 (IL-18Rα, IL-1Rrp, CD218a) Q13478	IL-1R8 (TIR8, SIGIRR) A0A291NLA3	Anti	None reported		
IL-36	IL-36α (IL-1F6) Q9UHA7	IL-36β (IL-1F8) Q9NZH7		IL-36γ (IL-1F9) Q9NZH8	IL-1R6 (IL-36R, IL-1Rrp2, IL-1RL2) Q9HB29	IL-1R3 (IL-1RAcP) Q9NPH3	Pro	IL-36Ra (IL-1F5) Q9UBH0		
	IL-38 (IL-1F10) Q8WWZ1				IL-1R6 (IL-36R, IL-1Rrp2, IL-1RL2) Q9HB29 * Early evidence only	Not known	Anti	None reported		
NA	* Early evidence only regarding ligand candidates				IL-1R9 (IL1RAPL-2, TIGIRR-1) Q9NP60	None reported	* Early evidence only regarding possible functions	None reported		
					IL-1R10 (IL1RAPL-1, TIGIRR-2) Q9NZN1					

Protein name; (synonyms); UniProt ID. Subfamily assignment by structural similarity. NA, not applicable.

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
