# Peer review of "The Role of the Interleukin-1 Family in Complications of Prematurity"

_ijms, 2023, doi:10.3390/ijms24032795_

Round 1

Reviewer 1 Report

Review by Green et al., highlights the importance of IL1 family cytokines in prematurity. The general concept of the manuscript is novel/significant for the field and the review is well-written. However, I have a few minor suggestions.

1.     Author should make one more column of cytokine subfamily and arrange them based on that classification in the table mainly because in text from line number 57 to 221  authors have arranged and described them based on subfamily classification but that is not very clear in the table.

2.     Line 652 to 653, there are several methods with minor changes to induce NEC in mice. Please cite - PMID: 34816129; and add a few sentences describing this procedure as well.

Author Response

  1. We appreciate Reviewer #1 pointing out that there was room for improvement in the presentation of the table layout. We have addressed the Reviewer’s suggestion and amended Table 1 to include a column detailing the cytokine subfamily.
  2. The lines 765 to 766 describe a specific protocol to induce NEC, the findings of which were detailed in the following sentence. Therefore, we have now included information in the introductory paragraph (lines 720 to 723) detailing how NEC can be induced experimentally in animals. In this explanation we now cite PMID: 34816129, in addition to a review on NEC models.

Reviewer 2 Report

This is a very interesting and well written review focused on the role of the Interleukin-1 family cytokines in complications of prematurity.

Only minor revisions are required. In particular:

Introduction: since this is a review article where preterm delivery is an important topic, the term "preterm birth (PTB)" should be added toghether with the term "premature ropture of membranes (PROM)". In fact, both these terms are not even mentioned but are important causes of prematurity. Moreover, it deserves to be specified that PROM is mainly due to infection (mainly chorioamnionitis) and inflammatory cytokines such as IL-1B (the topic of this review) that weaken placental membranes destroying tight junctions (as demonstrated in PMID: 26739007, 24768095). This is an important point to highlight because PROM occurs in about 30% of preterm delivery. 

Author Response

We thank Reviewer #2 for her/his constructive comments.

We agree that PROM is an important topic as highlighted by the Reviewer.  As such, we addressed this request by adding lines 291-293. We also agree that the role of IL-1 family members in the induction of preterm birth is indeed another important topic; however, induction of preterm birth itself is not the main focus of this review. Nonetheless, the revised version introduces the term “preterm birth” at the beginning of the Introduction and utilizes it many times throughout the text. Section 3 now titled “Preterm birth and sources of fetal inflammation” and comprises information about PROM.